# Identification of a Novel Mutation Exacerbated the PSI Photoinhibition in *pgr5*/*pgrl1* Mutants; Caution for Overestimation of the Phenotypes in Arabidopsis *pgr5-1* Mutant

**DOI:** 10.3390/cells10112884

**Published:** 2021-10-26

**Authors:** Shinya Wada, Katsumi Amako, Chikahiro Miyake

**Affiliations:** 1Graduate School of Agricultural Science, Kobe University, 1-1, Rokkodai, Nada-ku, Kobe 657-8501, Japan; cmiyake@hawk.kobe-u.ac.jp; 2Department of Health and Nutrition, Faculty of Human Life Studies, Jin-ai University, 3-1-1, Ohde-cho, Fukui, Echizen 915-8586, Japan; amakok@jindai.ac.jp; 3Core Research for Environmental Science and Technology (CREST), Japan Science and Technology Agency (JST), 7, Gobancho, Chiyoda-ku, Tokyo 102-0076, Japan

**Keywords:** proton gradient regulation 5 (PGR5), PGR5-like photosynthetic phenotype 1 (PGRL1), photosynthetic electron transport, PSI photoinhibition, oxidation of P700, oxidative stress

## Abstract

PSI photoinhibition is usually avoided through P700 oxidation. Without this protective mechanism, excess light represents a potentially lethal threat to plants. PGR5 is suggested to be a major component of cyclic electron transport around PSI and is important for P700 oxidation in angiosperms. The known Arabidopsis PGR5 deficient mutant, *pgr5-1*, is incapable of P700 oxidation regulation and has been used in numerous photosynthetic studies. However, here it was revealed that *pgr5-1* was a double mutant with exaggerated PSI photoinhibition. *pgr5-1* significantly reduced growth compared to the newly isolated PGR5 deficient mutant, *pgr5^hope1^*. The introduction of PGR5 into *pgr5-1* restored P700 oxidation regulation, but remained a pale-green phenotype, indicating that *pgr5-1* had additional mutations. Both *pgr5-1* and *pgr5^hope1^* tended to cause PSI photoinhibition by excess light, but *pgr5-1* exhibited an enhanced reduction in PSI activity. Introducing AT2G17240, a candidate gene for the second mutation into *pgr5-1* restored the pale-green phenotype and partially restored PSI activity. Furthermore, a deficient mutant of PGRL1 complexing with PGR5 significantly reduced PSI activity in the double-deficient mutant with AT2G17240. From these results, we concluded that AT2G17240, named PSI photoprotection 1 (PTP1), played a role in PSI photoprotection, especially in PGR5/PGRL1 deficient mutants.

## 1. Introduction

Photosynthesis consists of two steps: the electron transport reaction and the carbon fixing reactions. The electron transport reaction converts light energy absorbed in chloroplast thylakoid membrane to chemical energy (NADPH and ATP), while the subsequent carbon fixing reaction (Calvin–Benson cycle) consumes NADPH and ATP to fix CO_2_. These reactions are regarded as an electron source-sink relationship. The electron transport reaction consists of photophysical and biochemical processes, while the carbon fixing reaction is biochemical; therefore, the impacts of environmental stresses (such as strong light, temperature, drought, etc.) are expected to be different between these reactions, despite the activities of the two reactions being tightly linked [1].

An imbalance between electron source and sink can cause fatal damage to the photosynthetic apparatus, especially on photosystem I (PSI). When the electron transport chain is full of electrons, O_2_ can be easily reduced into the superoxide radical (O_2_^−^) on the components with the lowest redox potential, in other words, the acceptor side of PSI [2]. Sejima et al. (2014), [3], devised an experimental method called repetitive short-pulse (rSP) illumination that promoted the electron accumulation in the electron transport chain without activation of the carbon fixing reaction. This rSP-illumination drives the electron source while suppressing the drive of the electron sink. That is, rSP-illumination induces the imbalance of the electron accumulation between them. Consequently, rSP-illumination induced O_2_^−^dependent PSI-specific inactivation. The acceptor side of PSI is the main generation site for reactive oxygen species (ROS) which cause oxidative damage to PSI (PSI photoinhibition) [2,4,5]. Photo-inactivated PSI has been shown to take days or weeks for full recovery [6,7,8,9], causing severe reductions in CO_2_ fixation rate [3,10,11]. Thus, PSI photoinhibition can severely affect growth and may even be lethal for plants.

However, in nature, PSI photoinhibition rarely occurs in wild plants [12]. Under unsuitable environments for photosynthesis, coordinated linkage of the electron source and sink promotes oxidation of the PSI reaction center, P700, avoiding PSI photoinhibition [1,3,4]. P700 transports electrons as part of a redox cycle with three energy states. The oxidized form, P700^+^, receives electrons from PSII via plastocyanin (PC) and is reduced into P700. The reduced form P700 is photo excited to P700*. P700* then passes the electron to the next electron acceptor and is re-oxidized to P700^+^. The increase of oxidized P700 suggests that the rate-determining step in the P700 redox cycle is the reduction of P700^+^ [1]. Under stress conditions, where CO_2_ fixation (electron sink) was suppressed, P700 was generally more oxidized [13,14,15,16,17]. Theoretically, a decrease in the electron sink would predict a significant reduction of P700 (electron source). However, at this time, the trans-thylakoid ΔpH is increased, suppressing the electron transport activity of Cyt *b6f* [18]. Additionally, the redox balance of the plastoquinone (PQ) pool also slows the electron flow at the Cyt b6f [19]. The extent of the suppression of electron flow to PSI was larger than that of the decrease in the electron sink, resulting in oxidized P700 more [1,20]. Conversely, under the specific conditions (chilling and rSP-illumination) where PSI photoinhibition was observed, P700 oxidation was not induced [3,9]. PSI photoinhibition has also been observed in some mutants that could not promote P700 oxidation [21,22]. Oxidation of P700 avoids the accumulation of electrons in the acceptor side of PSI where ROS are generated, while oxidized P700 cannot use the light energy absorbed in PSI for electron transport but can directly dissipate it as heat [23,24,25]. In plants, promoting P700 oxidation can be regarded as a robust mechanism of avoiding ROS generation and protecting PSI.

Proton gradient regulation 5 (PGR5), a protein tethered in the thylakoid membrane, was identified in an Arabidopsis deficient mutant, *pgr5* [21], renamed as *pgr5-1* [26]; hereafter, also referred to as “*pgr5-1*” in this study. *pgr5-1* reduced proton gradient formation and P700 oxidation under high light [21,27]. PGR5 complexing with PGRL1 has been proposed to be a major component in cyclic electron flow around PSI [28]; however, the molecular function of PGR5 remains unclear. Under high light, preferential PSI photoinhibition was often observed in *pgr5-1* [21,26,29,30,31], and it was assumed that ROS production was accelerated by the accumulation of reduced iron-sulfur centers due to the loss of P700 oxidation. Moreover, *pgr5-1* significantly reduced survivability and growth under natural light conditions [32], indicating that PSI photoprotection by P700 oxidation is necessary for acclimation to natural light. Therefore, *pgr5-1* has been used as a valuable tool to investigate PSI photoinhibition, possible lethal damage caused by natural light stress [31,32,33].

In this study, we newly isolated a PGR5 deficient mutant named *pgr5^hope1^*. Coincidentally, *pgr5^hope1^* and the known *pgr5-1* mutant were revealed as having the same point mutation in *PGR5* gene, although, *pgr5-1* grew smaller and had lower photo-oxidizable PSI (*Pm*) than *pgr5^hope1^* under both constant and natural light conditions. The phenotypic differences suggested the presence of additional factors besides PGR5, which was assumed to be related to the PSI photoprotection. So far, little is known about the factors involved in PSI photoprotection. Therefore, we aimed to find the novel factor, which we named PSI photoprotection 1 (PTP1). Whole-genome sequencing and complemented transformants revealed that PTP1 was encoded in AT2G17240 gene. AT2G17240 gene was identified previously as cgl20a, which interacted with plastidial ribosomes and affected plastome translation [34]. A mutation of the *PTP1* gene in *pgr5-1* enhanced PSI photoinhibition and exacerbated the growth reduction. Here, we introduce the novel factor involved in PSI photoprotection in Arabidopsis.

## 2. Materials and Methods

### 2.1. Plant Materials and Growth Condition

*Arabidopsis thaliana* (L.) Heynh. ecotype Columbia (Col) gl-1 and Col-0 were used as wild-type. In mutants used in this study, *pgr5^hope1^* and *pgr5-1* were gl-1 backgrounds, and *ptp1-1*, *npq4* and *pgrl1ab* were Col-0 backgrounds. *pgr5^hope1^* was screened from an ethyl methanesulfonate (EMS) mutant population, which was gl-1 background, by chlorophyll fluorescence measurement under low O_2_ and CO_2_ free conditions (Appendix A) [35]. *pgr5-1, ptp1-1, npq4* and *pgrl1ab* were originated from [21], [34], [36], and [22], respectively. For the constant light condition, plants were cultivated in a growth chamber (BioTRON LPH-241, NKsystem, Osaka, Japan) with a light/dark regime of 10/14 h (light intensity; 250 μmol photons m^−2^ s^−1^) with the temperature of 24/22 °C. For natural light conditions, plants were cultivated in a greenhouse with supplemental halogen lamps used in the morning and evening to extend the day length to 14 h and adjusted temperature at 23/20 °C, during March–May in 2020. The greenhouse is located in Kobe University (34°43′ north latitude, 135°14′ east longitude). Based on the climate data in Kobe, the daytime light intensity varied up to around 2000 μmol photons m^−2^ s^−1^. The culture soil was a mixture of vermiculite and horticultural soil (Tanemaki-baido, Takii, Kyoto, Japan) at a ratio of 1:1.

### 2.2. Transformation

To produce complementation lines, *pgr5-1 PTP1, pgr5-1 PGR5* and *pgr5^hope1^ PGR5* constructs expressing *PGR5* or *PTP1* cDNA under the control of 35S promoter were introduced into *pgr5-1* or *pgr5^hope1^* by agrobacterium-mediated transformation (*Agrobacterium tumefaciens* strain GV3101). mRNA was isolated from leaves of wild-type gl-1 using RNeasy plant mini kit (QUIAGEN, Hilden, Germany), which was used for cDNA synthesis by PrimeScript RT master mix (Takara, Shiga, Japan). Full-length CDS of *PGR5* and *PTP1* was amplified from the cDNA with specific primers described in Appendix A. Amplified CDSs were primarily inserted into the pENTR/D-topo entry vector (Thermo Fisher Scientific, Waltham, MA, USA) and finally introduced into pBI DAVL-GWR1 destination vectors (Inplanta Innovations, Yokohama, Japan) by the Gateway cloning system. Transformation into plants was performed by floral dip methods [37].

### 2.3. Determination of Phenotypic Recovery in Complementation Lines

The phenotypic recovery was assessed by four different parameters, the shoot fresh weight, SPAD value, the maximum quantum yield of PSII (*Fv/Fm*), and the maximum amount of photo-oxidizable P700 (PSI) (*Pm*). Sampled plants were weighed and the SPAD values were measured using a SPAD-502 (Konica Minolta, Tokyo, Japan). *Fv/Fm* and *Pm* were measured by Dual-Pam/F (Walz, Effertrich, Germany). The calculations of *Fv/Fm* and *Pm* were described below.

### 2.4. Simultaneous Measurement of Gas Exchange, Chlorophyll Fluorescence, and Absorbance Change Due to P700 Oxidation

Analysis of photosynthesis was conducted on rosette leaves of Arabidopsis about 30 days after sowing. CO_2_ and H_2_O gas exchanges were measured by a GFS-3000 system equipped with a Dual-PAM gas-exchange cuvette (Walz, Effertrich, Germany). For the measurement, ambient air CO_2_ concentration (Ca), relative humidity, and leaf temperature were controlled at 400 ppm, 60%, and 25 °C, respectively. CO_2_ fixation rate (*A*) was calculated by the system software based on the method of [38].

Chlorophyll fluorescence and absorbance change were measured using Dual-PAM 100 (Walz) simultaneously with gas exchange. Chlorophyll fluorescence parameters were calculated as follows based on [39]; *Fo*, minimal fluorescence in dark-adapted leaf; *Fm*, maximal fluorescence in dark-adapted leaf; *Fm’*, maximal fluorescence in light-adapted leaf; *Fs*, fluorescence in steady-state; the effective PSII quantum yield (Y(II)), Y(II) = (*Fm’* − *Fs*)/*Fm’*; non-photochemical quenching (NPQ), NPQ = (*Fm* − *Fm’*)/*Fm’*. And P700 absorbance parameters were calculated as follows based on [15]; *Pm*, the maximal signal of photo-oxidizable P700; *Pm’*, the maximal signal of P700 photo-oxidized by saturating pulse flash under actinic light; *P*, the signal of P700 photo-oxidized under actinic light; the effective PSI quantum yield (Y(I)), Y(I) = (*Pm’* − *P*)/*Pm*; the quantum yield of non-photochemical energy dissipation due to donor-side limitation of PSI (Y(ND)), Y(ND) = *P*/*Pm*; the quantum yield of non-photochemical energy dissipation due to acceptor-side limitation of PSI (Y(NA)), Y(NA) = (*Pm* − *Pm’*)/*Pm*. The sum of these quantum yields is 1. (Y(I) + Y(ND) + Y(NA) = 1) To determine the *F*o and *F*m, fully dark-adapted leaves (<30 min) were irradiated with a saturated light pulse (20,000 μmol-photons m^−2^ s^−1^, for 300 ms). The maximal quantum yield of PSII, *Fv*/*Fm*, was calculated as; *Fv*/*Fm* = (*Fm* − *Fo*)/*Fm*. After *Fo* and *Fm* determination, *Pm* was determined by a saturating pulse in the presence of far-red light. Then, leaves were irradiated with actinic light. The light intensity of actinic light was increased stepwise (8.2, 26.1, 54.8, 110.8, 200.3, 434.6, 709.3, 903.0, 1150.3, 1447.6 μmol-photons m^−2^ s^−1^), and the respective parameters were measured at each light intensity with application of a saturating pulse. After the photosynthesis measurement, the leaves were excised, frozen with liquid nitrogen, and stored at −80 °C before biochemical analysis.

### 2.5. Biochemical Analyses

The leaf N, chlorophyll and rubisco protein content were determined according to [40]. Leaf samples were ground with homogenization buffer (50 mM sodium phosphate, 5%(*v*/*v*) glycerol and 1 mM sodium iodoacetate). The aliquot was used for leaf N and chlorophyll determination based on the Kjeldahl method and [41]. For rubisco protein quantification, leaf soluble fractions were applied to sodium dodecyl sulfate polyacrylamide gel electrophoresis (SDS-PAGE). After Coomassie brilliant blue (CBB) staining, the bands corresponding to the large and small subunits of rubisco were excised, and the dye extracted with formamide was colorimetrically quantified. Bovine serum albumin (BSA) protein (Bovine Serum Albumin Standard, Thermo Fisher Scientific, Waltham, MA, USA) was used as a standard sample, and a standard curve was prepared.

### 2.6. Immunoblotting

Aliquots of leaf homogenate for biochemical analyses were applied for immunoblotting. The total leaf homogenate was combined with an equal volume of SDS-sample buffer (200 mM Tris-HCl (pH 8.5), 2%(*w*/*v*) SDS, 20%(*v*/*v*) glycerol, 5%(*v*/*v*) 2-ME), boiled for 2 min and stored at −30 °C until analysis. Immunoblotting was carried out with a 12.5%(*w*/*v*) acrylamide gel, a semi-dry blotting apparatus (Trans-Blot Turbo Transfer System; Bio-Rad, Hercules, CA, USA), a polyvinyldifluoridene (PVDF) membrane (Trans-Blot Turbo RTA transfer kit, mini, PVDF; Bio-Rad), a chemiluminescence detection kit (SuperSignal West Dura Extended Duration Substrate; Thermo Fisher Scientific), and an image analyzer (Ex-Capture MG; ATTO, Tokyo, Japan). All antibodies used in this study were purchased from Agrisera (Vännäs, Sweden). The product code of PsaA, Lhca1, PsbA, Lhcb5, PETB, NDHB and NDHH specific antibodies were AS06 172, AS01 005, AS05 084A, AS01 009, AS18 4169, AS16 4064 and AS16 4065, respectively. The Goat Anti-Rabbit IgG Horseradish Peroxidase Conjugated (Thermo Fisher Scientific) was used for the secondary antibody.

### 2.7. Photoinhibition Experiment

For the photoinhibition in PSII and PSI, attached leaves of plants were exposed to actinic light (900 μmol-photons m^−2^ s^−1^) for 2 h in a Dual-PAM cuvette and Dual-PAM 100 (Walz, Effertrich, Germany). The experimental condition was the same as in the measurement of photosynthesis parameters described above. *Fv*/*Fm* and *Pm* were determined before tuning on the actinic light and after 30 min dark treatment after turning off the light.

### 2.8. Whole-genome Sequences Analysis

The whole-genome sequence analysis was performed with NovaSeq 6000 by Macrogen (Kyoto, Japan). The TAIR 10.1 was used as a reference genome sequence of Arabidopsis.

## 3. Results

### 3.1. Identification of the PSI Photoprotection 1, PTP1; a Recessive Mutation in AT2G17240 Gene Enhanced the Growth Phenotype of pgr5-1, a Well-known PGR5 Deficient Mutant

We previously performed screening for Arabidopsis mutants in ΔpH formation across thylakoid membrane by monitoring chlorophyll fluorescence under hypoxic conditions (*hope* mutant screening; hunger for oxygen in photosynthetic electron transport reaction) [34]. *hope1* was isolated as a high chlorophyll fluorescent mutant in hypoxic conditions (Appendix A) and was identified a mutation in proton gradient regulation 5 (*PGR5)* gene by genome mapping. Coincidently, the mutation was in the same position (388 G to A) as *pgr5-1*, a known PGR5 deficient mutant (Appendix A [21]). The mutation resulted in PGR5 protein deficiency in both *pgr5-1* and *hope1* (Appendix A). As *pgr5-1* and *hope1* were genetically the same alleles as *pgr5* mutants, hereafter, *hope1* was referred to as *pgr5^hope1^* in this study.

Figure 1 showed the growth phenotypes of *pgr5-1* and *pgr5^hope1^* grown under relatively high (250 μmol-photons m^−2^ s^−1^) constant light (Figure 1A,B), and natural light condition (varying below 2000 μmol-photons m^−2^ s^−1^; Figure 1C,D). Despite the identical mutation, under both light conditions, *pgr5-1* was significantly smaller than *pgr5^hope1^*. Although the growth of *pgr5-1* was severely affected by light intensity, the growth declines of *pgr5-1* under constant and natural light conditions were consistent with [32,42]. Whereas *pgr5^hope1^* showed almost the same growth as wild-type under constant light conditions, although it tended to be slightly smaller (Figure 1A,B). Under natural light, *pgr5^hope1^* exhibited a reduction to around 35% of the fresh weight compared to the wild-type, while *pgr5-1* achieved masses around 11% of wild-type (Figure 1C,D). These differences suggested that the influence of genetic factors other than *PGR5* were involved in these growth phenotypes. All F1 hybrids of *pgr5-1* and *pgr5^hope1^* grew the same level as *pgr5^hope1^* (Figure 1A,B). Based on the fresh weight and the SPAD value, an indicator for chlorophyll amount, the growth phenotype of the F2 progeny exhibited Mendelian segregation of approximately 1:3 (16:62) in 78 plants, indicating the existence of a second genetic factor affecting *pgr5-1* or *pgr5^hope1^* growth. The growth reduction in *pgr5-1* compared to the wild-type was reported to be mainly due to the PSI photoinhibition [21,43]. We considered that the second mutation could act positively or negatively on the PSI photoinhibition in *pgr5* mutant, resulting in different phenotypes. Thus, we named this factor PSI photoprotection 1, *PTP1*.

For genetic identification, we performed whole-genome sequencing on *pgr5-1* and *pgr5^hope1^*. So far, *pgr5-1* and *pgr5^hope1^* have no phenotypic segregation in hybrid strain production or backcrossing. Therefore, it was considered that the *PTP1* was positioned near the *PGR5* locus on chromosome 2 and was difficult to segregate from PGR5 mutation. The candidate mutations on chromosome 2 that existed independently in each of these mutants and that had a severe effect (e.g., amino acid substitution) were listed in Appendix A. We selected several candidate mutations from the list and analyzed their T-DNA insertion mutants. One of them, *salk_133989*, a T-DNA insertion allele for AT2G17240, exhibited a pale green phenotype which partially resembled *pgr5-1* phenotype (Appendix A). AT2G17240 was reported to encode CGL20A protein and the *salk_133989* was its defective mutant [34]. A point mutation in AT2G17240 was found in *pgr5-1* (Appendix A) which altered C to T at base 278 of the coding DNA, predicting an amino acid substitution at position 93 from Pro to Leu (Appendix A). Here, we tentatively termed AT2G17240 gene *PTP1*, while the mutant alleles, T-DNA insertion mutant alleles (*salk_133989*) and one base substitution in *pgr5-1* were named *ptp1-1* and *ptp1-2*, respectively (Appendix A).

To determine whether PTP1 caused the phenotypic difference between *pgr5^hope1^* and *pgr5-1*, we produced several complementation lines and observed phenotypic recoveries under constant (Figure 2A–C) and natural light condition (Figure 2D–F). We assessed the phenotypic recovery as fresh weight and SPAD value (Figure 2B,E), and the maximum yield of PSII (*Fv*/*Fm*) and the maximum amount of photo-oxidizable PSI (*Pm*; Figure 2C,F). The genetic background is different between *pgr5-1*, *pgr5^hope1^* (gl-1) and *ptp1-1* (Col-0). However, the wild-type gl-1 and Col-0 didn’t show any significant differences in their growth (Figure 1A,D) and measured parameters (Figure 1B,C,E,F). Under the constant light condition, *pgr5^hope1^* slightly reduced the fresh weight, SPAD value, and *Pm* compared to wild-type (Figure 2B,C). The complementation line, *pgr5^hope1^ PGR5*, which induced *PGR5* in the *pgr5^hope1^* background under the control of 35S promoter, restored these reductions to the wild-type level (Figure 2B,C). *pgr5^hope1^* significantly decreased in growth and *Pm* under natural light, but *pgr5^hope1^ PGR5* restored these decreases to the wild-type level (Figure 2E,F). These results strongly indicated that *pgr5^hope1^* was a *pgr5* single mutant. In contrast, *pgr5-1* significantly reduced fresh weight, SPAD value, and *Pm* under both constant and natural light conditions (Figure 2B,C,E,F). *pgr5-1 PGR5*, which induced *PGR5* in the *pgr5-1* background, largely restored growth rate and *Pm*. However, *pgr5-1 PGR5* remained low SPAD value and showed pale green leaves similar to *ptp1-1*. Partial phenotypic recovery of *pgr5-1 PGR5* suggested the existence of other mutations causing the pale green leaves. Whereas *pgr5-1 PTP1*, which expressed *PTP1* in *pgr5-1* background, restored the fresh weight, the SPAD value and *Pm* to almost the same level as *pgr5^hope1^* under constant light condition (Figure 2B,C). In addition to constant light, *pgr5-1 PTP1* restored these parameters to the *pgr5^hope1^* level under natural light (Figure 2E,F). These recoveries indicated that the introduction of *PTP1* in *pgr5-1* was largely compensated for the phenotypic differences between *pgr5-1* and *pgr5^hope1^*.

To confirm the effect of double mutation of *PTP1* and *PGR5*, we attempted to crossbreed *pgr5^hope1^* and *ptp1-1*. However, the double mutant could not be obtained from the cross-bred strains, probably due to the proximity of their loci on chromosome 2. Alternatively, we produced a double-deficient mutant of *PGRL1* and *PTP1* (*pgrl1ab ptp1-1*) and analyzed the growth phenotypes under constant light conditions (Figure 3). PGRL1 forms a protein complex with PGR5, and the deficient mutant, *pgrl1ab*, is incapable of P700 oxidation like PGR5 deficient mutant [22]. *pgrl1ab ptp1-1* double mutant significantly reduced the growth and *Pm* like *pgr5-1*, while the *pgrl1ab* grew similar to wild-type like *pgr5^hope1^* (Figure 3). In addition, we produced a double mutant *npq4 ptp1-1* to compare the effect of PTP1 deficiency in another photodamage susceptible mutant. *npq4* mutant is deficient in PsbS protein, reducing its ability of photoprotective thermal energy dissipation [36]. Consequently, the growth, *Fv*/*Fm* and *Pm* did not change between *npq4* and *npq4 ptp1-1* (Figure 3), indicated that PTP1 deficiency did not exacerbate the growth inhibition of *npq4*. From the results of complementation lines and *pgrl1ab ptp1-1*, we concluded that the second genetic factor *PTP1* affecting the growth phenotype in *pgr5-1* was AT2G17240 gene. As a result, *pgr5-1* was determined to be a double mutant of *pgr5-1 ptp1-2*. We tried to but have been so far unsuccessful in detecting the PTP1 protein, even in wild-type plants. Therefore, the detailed effect of the *ptp1-2* mutation on PTP1 protein in *pgr5-1* remained unclear. However, under constant light conditions, *pgr5-1 PTP1* and *ptp1-1* deficient mutant reduced the SPAD value to a similar level (Figure 2B,E) and reduced chlorophyll content to the same level (Table 1). Furthermore, in comparison with wild-type, the growth reduction in *pgr5-1* was relatively larger than in the *pgrl1ab ptp1-1* (Figure 2B and Figure 3B). Thus, it was estimated that the *ptp1-2* mutation in *pgr5-1* caused a similar situation to protein deficiency or further impairment of its function.

### 3.2. The Biochemical and Physiological Damages on Photosynthetic Apparatus Caused by the ptp1 Mutation

*pgr* allele possessed the second recessive mutation, *ptp1-2*, and was regarded as a double mutant, *pgr5-1 ptp1-2*, whereas *pgr5^hope1^* was regarded as a single PGR5 deficient allele, based on the phenotypic recovery. To reveal the molecular function of PTP1, we compared the difference between *pgr5^hope1^* and *pgr5-1*. First, we determined the total leaf N, chlorophyll and rubisco content (Table 1), and thylakoid protein levels, mainly in electron transport chain components of plants grown under constant light conditions (Figure 4). No significant differences in leaf total N and rubisco content were noted between genotypes. On the other hand, *pgr5-1* and *ptp1-1* deficient mutants exhibited significantly reduced chlorophyll contents compared to wild-type and *pgr5^hope1^*; consistent with the SPAD value and the pale green leaf color (Figure 1A and Figure 2A). But the chlorophyll a/b ratio was not significantly changed in *pgr5-1* and *ptp1-1* (Table 1). Moreover, only *pgr5-1* largely reduced the concentration of the core subunit of PSI (PsaA) to around 66% of wild-type levels, but not other tested proteins in the electron transport chain (Lhca1, PsbA, Lhcb5, and PETB). Previous studies have shown that the core subunit of PSI, PsaA, is specifically degraded in the process of PSI photoinhibition [9,43]. In *pgr5-1*, the specific decrease of the PSI core protein was observed with PSI inactivation [32,44], suggesting severe PSI photoinhibition. However, the PsaA level in *pgr5^hope1^* was only marginally reduced, to around 90% of wild-type levels (Figure 4). Thus, the PSI photoinhibition in *pgr5-1* was suggested not only due to PGR5 deficiency but was enhanced by the *ptp1-2* mutation. In addition, *ptp1-1* deficient mutant was reported a decrease in proteins of NDH complex [34]. We also detected the decrease of NDHB and NDHH proteins in the complex to around 70% of wild-type in *ptp1-1* and *pgr5-1* (Figure 4). The growth reduction of *pgr5-1* was largely enhanced by crossing with NDH complex deficient mutants (*crr2* and *crr4*) [45]. The reduction of NDH complex by PTP1 mutation may be associated with the growth phenotype of *pgr5-1* and *pgr5^hope1^*.

PSI photoinhibition has been shown to cause severe reductions in CO_2_ fixation rate [3,10,11]. We next determined CO_2_ fixation rate (*A*), chlorophyll fluorescence and absorbance change simultaneously, using plants grown under constant light conditions (Figure 5). *A* in *pgr5-1* was significantly reduced, especially under high light (<500 μmol-photons m^−2^ s^−1^), to about 60% of wild-type (Figure 5A). A large reduction of *A* in *pgr5-1* was previously observed in [42], mainly due to PSI photoinhibition caused by PGR5 deficiency. *pgr5^hope1^* also reduced *A*, but not as much as *pgr5-1*, to about 90% of wild-type levels under high light (<500 μmol-photons m^−2^ s^−1^) condition. This difference in *A* between *pgr5^hope1^* and *pgr5-1* suggested that the *PTP1* mutation largely enhanced the reduction in *A* under PGR5 deficient condition. A similar decrease in *A* was confirmed between *pgrl1ab ptp1-1* and *pgrl1ab* (Appendix A). In contrast, both *pgr5-1* and *pgr5^hope1^* exhibited similarly reduced quantum yields in PSII and PSI, Y(II) and Y(I), compared to wild-type (Figure 5B,D). *pgr5-1* was originally isolated as a mutant incapable of qE-dependent non-photochemical quenching (NPQ), an indicator for trans-thylakoid ΔpH formation, and P700 oxidation [Y(ND)] under high light ([21]; Figure 5C,E in this study). *pgr5^hope1^* also showed the same low levels of NPQ and Y(ND) as *pgr5-1*. These results indicated that changes in electron transport capacities, including the loss of P700 oxidation, were due solely to PGR5 deficiency. In contrast, *ptp1-1* deficient mutants showed almost the same photosynthetic capacities as wild-type (Figure 5). These results indicated that PTP1 was neither necessary in photosynthesis nor directly involved in P700 oxidation.

### 3.3. PSI Photoinhibition Was Triggered by PGR5/PGRL1 Deficiency

The specific core protein degradation in PSI and the reduction in CO_2_ fixation rate suggested that the mutation of PTP1 enhanced PSI photoinhibition in *pgr5-1* (Figure 4 and Figure 5A). Next, we analyzed the extent of photoinhibition in PSII and PSI caused by constant intense light (900 μmol-photons m^−2^ s^−1^), using plants grown under constant light conditions (Figure 6). In wild-type and *ptp1-1*, no changes in the maximum quantum yield of PSII (*Fv*/*Fm*) and the maximum amount of photo-oxidizable PSI (*Pm*) were found before or after high light irradiation (Figure 6B,C). It suggested that PTP1 deficiency was not a trigger for PSI photoinhibition. On the contrary, *pgr5^hope1^* and *pgr5-1* significantly decreased both *Fv*/*Fm* and *Pm*, especially *Pm*, after irradiation. Similar results were observed with the *pgrl1ab* mutant (Appendix A). These results showed that PSI photoinhibition, whether PTP1 was present or absent, was primarily induced by PGR5/PGRL1 deficiency. The reduction rate of *Pm* was significantly larger in *pgr5-1* (53%) than in *pgr5^hope1^* (40%). However, *pgr5-1* exhibited significantly lower *Pm* before irradiation, suggesting that the actual difference in PSI damage was even smaller. The low *Pm* before high light irradiation also suggested that the damage of PSI photoinhibition was accumulated constantly in *pgr5-1*. These results imply that PTP1 did not avoid the occurrence of PSI photoinhibition but suppressed the damages to PSI.

## 4. Discussion

In this study, we identified the AT2G17240 gene, named *PTP1*, as a novel factor for PSI photoprotection. Mutations in PTP1 exacerbated PSI photoinhibition and reduced the growth of *pgr5* and *pgrl1* mutants. PGR5 and PGRL1 that form a protein complex are suggested to be the main components of cyclic electron transport around PSI (CEF-PSI) [21,22], but its actual molecular function has not yet been specified. *ptp1-1* deficient mutant performed the same photosynthetic activity including ΔpH formation (NPQ) as wild-type plants (Figure 5). In addition, PTP1 was reported as CGL20A to function in ribosome biogenesis in plastids [34]. Thus, PTP1 was unlikely to be directly involved in the linear electron transport or CEF-PSI. The physiological roles of CEF-PSI were suggested to protect PSI from excess light energy by sustaining high ΔpH and produce ATP for CO_2_ assimilation [28]. However, *pgr5^hope1^* showed significantly larger *Pm* than *pgr5-1* under both constant and natural light conditions (Figure 2C,F) and showed nearly the same CO_2_ assimilation level as wild-type under saturated light conditions (Figure 5A). These results indicated that the physiological roles of CEF-PSI based on the analysis of *pgr5-1* could have been overestimated.

In addition, it was suggested that the existence of CEF-PSI itself needed to be reconsidered in other experiments. As for the CEF-PSI pathway carried by PGR5 and PGRL1, ferredoxin-quinone reductase dependent (FQR) pathway in which electrons transfer from ferredoxin (Fd) to plastoquinone (PQ) in the electron transport chain is proposed. However, the electron transport rates at Fd and PSII showed a positive linear relationship without intercept, suggesting that electron transports other than the linear electron transport were negligible in abundance [46]. Furthermore, a direct measurement system of CEF-PSI has not been established so far, and the difference in quantum yield between PSI and PSII (ΔY(I) = Y(I) − Y(II)), which has been used as an alternative index, is also doubtful as an evaluation of CEF-PSI. The quantum yield of PSI (Y(I)) at steady state measured by saturation pulse irradiation could be overestimated depending on the redox state of plastocyanin (PC) [20]. Assuming that ΔY(I) was properly assessing the CEF-PSI, ΔY(I) was not induced by PGR5/PGRL1 deficient mutants not only under steady-state but also under fluctuating light [26]. However, the introduction of moss flavodiiron protein (FLV) into *pgr5-1* mutant restored ΔY(I) along with the oxidation regulation of P700 [26]. This recovery was further confirmed by the introduction of FLV in *PGR5-RNAi* transformant in rice (*Oryza sativa*) [47]. FLV, theoretically, played a role as an alternative electron sink for linear electron transport by transporting electrons to oxygen to generate water (pseudo-cyclic electron transport). However, FLV stimulated the photosynthetic linear electron transport and oxidized P700 in PSI, which induced the extra electron flux in PSI, ΔY(I), in both wild-type and *pgr5-1* plants [26]. These studies indicated that ΔY(I) was not induced by PGR5/PGRL1. In this study, ΔY(I) was also observed in wild-type, and not in *pgr5^hope1^* and *pgr5-1* (Figure 5B,C; Appendix A). However, the reduction of CO_2_ fixation rate in *pgr5-1* was caused by *ptp1-2* mutation, and *pgr5^hope1^* and wild-type exhibited no significant differences in CO_2_ fixation rate (Figure 5A). Therefore, it was shown that PGR5-dependent ΔY(I) contributed little to the CO_2_ fixation reaction even when evaluated functionally. On the other hand, the light stress sensitivity in *pgr5^hope1^* indicated that PGR5 was necessary for P700 oxidation regulation, and important for the avoidance of PSI photoinhibition (Figure 6). However, considering the growth of *pgr5^hope1^* compared to *pgr5-1* under constant light and natural light of some intensity, it was implied that there was a mechanism that suppressed PSI photoinhibition other than P700 oxidation regulation.

In the Arabidopsis genome, two highly homologous genes encode *CGL20*, *CGL20A* (PTP1; AT2G17240) and *CGL20B* (AT3G24506). It is interesting to see if these two genes duplicate the role of PSI photoprotection. The double deficient mutant, *cgl20ab*, exhibited significantly reduced plastome-encoded protein production, such as PnsL1-4 and PnSB1-2 in NDH complex, PsbA in PSII, PETB and PETD in Cyt *b6f*, PsaN in PSI [34]. Not only in the *cgl20ab* double knockout mutant, but *ptp1-1* (*cgl20a* single mutant) also showed a slight reduction in some subunits of NDH complex [34]. We also detected the decrease of subunits of NDH complex, NDHB and NDHH (Figure 4). NDH complex was Fd-dependent plastoquinone reductase that was proposed to be responsible for the minor pathway of CEF-PSI [48]. Although the mechanism was still unclear, *pgr5-1* was further reduced in growth by double deficiency with NDH complex [44]. The reduced NDH complex by the mutation of PTP1 may reduce the growth of *pgr5-1*. However, NDH deficient mutants, such as *crr2*, did not exhibit pale green phenotypes like *ptp1-1* [49]. The effect of *PTP1* mutation which enhanced PSI photoinhibition in PGR5/PGRL1 deficient mutants may have another cause within the chloroplast. Further investigation of the relationship between NDH complex and PSI photoinhibition under PGR5 deficiency may help to understand the molecular mechanisms of PSI photoprotection. On the other hand, *pgr5-1* significantly reduced PsaA level while *pgr5^hope1^* was nearly the same as wild-type (Figure 4). PSI protein turnover was relatively slow and full recovery is known to take several days [6,44,50]. In PSI, however, the core protein PsaA showed relatively faster turnover rate than other PSI proteins [51]. If PTP1 was responsible for PsaA turnover, it might alleviate PSI photoinhibition in its repair or reassembly process.

PSI photoinhibition is caused by ROS generated at the acceptor side of PSI [3,52,53]. ROS scavenging was considered as a photoprotective mechanism for photosystems [2]. In the acceptor side of PSI, some ROS scavenging enzymes, such as APX and SOD, function in the well-known alternative electron flow, the water–water cycle [5]. Arabidopsis has several APX and SOD genes all of which are encoded in the nuclear genome. Thus, PTP1 was unlikely to be involved in the expression of these proteins encoded in the nuclear genome. However, it was reported that chloroplastic Fe-SOD deficient mutant in Arabidopsis, *fsd2* and *fsd3*, showed sensitivity to oxidative stress [54,55], while in cyanobacteria, the lack of Fe-SOD strain caused the PSI photoinhibition [56]. A reduction in ROS scavenging capacities may enhance PSI photoinhibition.

## Figures and Tables

**Figure 1 cells-10-02884-f001:**
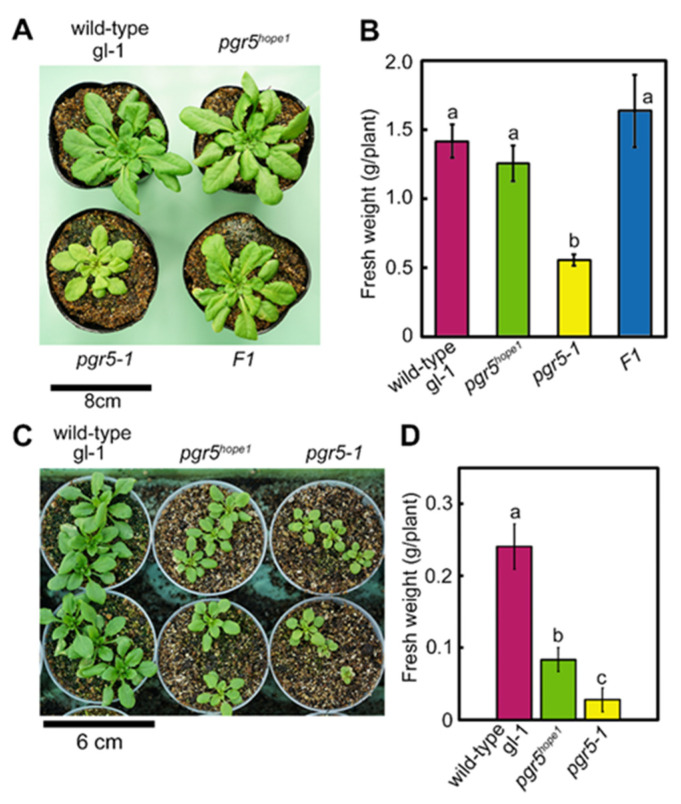
The difference of the growth phenotype between *pgr5-1* and *pgr5^hope1^*. (**A**) is a picture of representative plants (wild-type (gl-1), *pgr5^hope1^*, *pgr5-1*, and F1 hybrid of *pgr5-1* and *pgr5^hope1^*) grown under constant light conditions (250 μmol-photons m^−2^ s^−1^) for 30 days. (**B**) shows the fresh weight of plants in (**A**) Data are means ± sd. (*n* = 4–5) (**C**) is a picture of representative plants (wild-type (gl-1), *pgr5^hope1^* and *pgr5-1*) grown under natural light condition for 20 days. (**D**) shows the fresh weight of plants in (**C**). Data is means ± sd. (*n* = 9) In (**B**,**D**), Different alphabets indicate significant differences analyzed by Tukey’s HSD-test. (*p* < 0.05) Experiments were independently repeated at least 3 times and showed similar results. Figures showed the representatives.

**Figure 2 cells-10-02884-f002:**
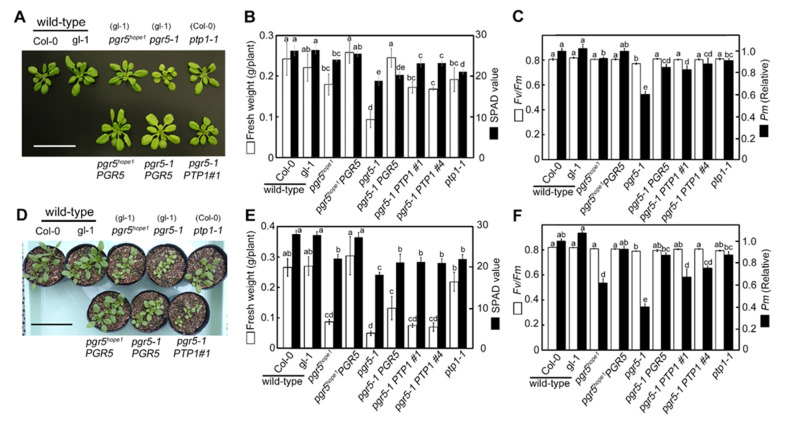
The phenotypic recovery in complementation lines. (**A**–**C**) and (**D**–**F**) were the data of plants grown under constant (250 μmol-photons m^−2^ s^−1^) and natural light conditions, respectively. (**A**,**D**) are pictures of representative plants grown for 22 days and 20 days, respectively. The parentheses indicate the background genotypes of each mutant. The scale bars in (**A**,**D**) are 5 and 8 cm, respectively. (**B**,**E**) show fresh weight (white bars) and SPAD values (black bars). Data are means ± sd. ((**B**), *n* = 6; (**E**), *n* = 4–5) (**C**,**F**) show the maximum quantum yield in PSII (*Fv/Fm*; white bars) and the maximum amount of photo-oxidizable PSI (*Pm*; black bars). Data are means ± sd. (C, *n* = 5–6; F, *n* = 4–5) *Pm* is a relative value with the value wild-type Col-0 being 1. In (**B**,**C**,**E**,**F**), the different alphabets indicate significant differences analyzed by Tukey’s HSD test. (*p* < 0.05).

**Figure 3 cells-10-02884-f003:**
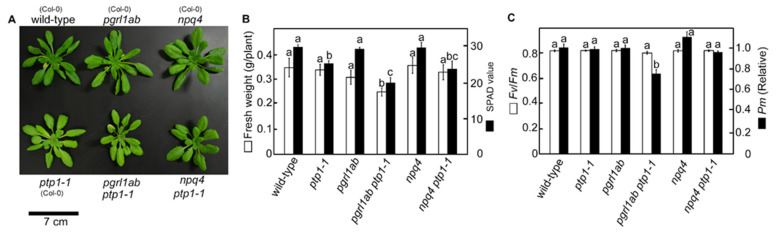
The effect of PTP1 mutation on the growth of PGRL1 deficient mutants. (**A–C**) were the data of plants grown under constant light conditions (250 μmol-photons m^−2^ s^−1^). **A** is a picture of representative plants grown for 28 days. The parentheses indicate the background genotypes of each mutant. The scale bar is 7 cm. (**B**) shows the fresh weight (white bars) and SPAD values (black bars). Data are means ± sd. (*n* = 4). (**C)** shows the maximum quantum yield in PSII (*Fv/Fm*; white bars) and the maximum amount of photo-oxidizable PSI (*Pm*; black bars). Data are means ± sd. (*n* = 4) *Pm* is a relative value with the value wild-type Col-0 being 1. In (**B**,**C**), the different alphabets indicate significant differences analyzed by Tukey’s HSD test. (*p* < 0.05).

**Figure 4 cells-10-02884-f004:**
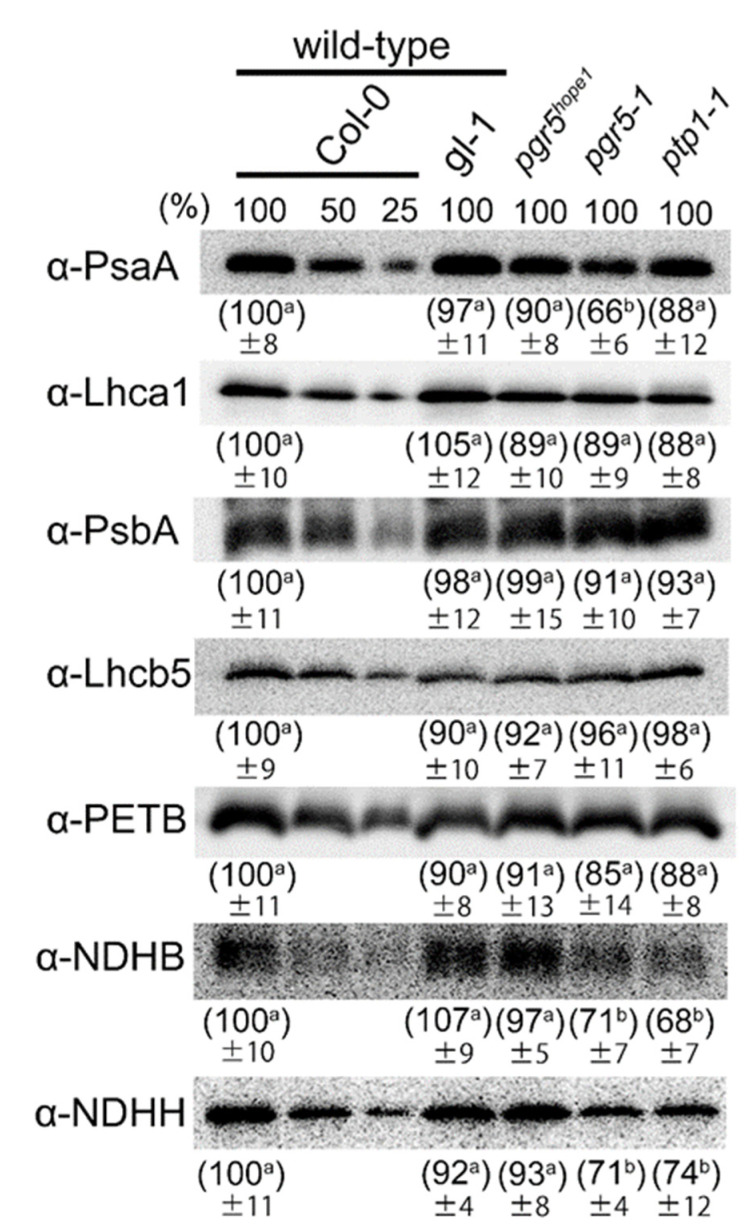
The immunoblotting of proteins in photosynthetic electron transport. Proteins in photosynthetic electron transport were detected with specific antibodies in each plant. Leaves were samples after measurements of photosynthetic parameters (Figure 5). Samples were loaded based on the same leaf area. The experiment was repeated at least three times with similar results. The values in parentheses show relative values of band intensities when the wild-type Col-0 is 100. Data are means ± sd. (*n* = 3–4) Different alphabets beside the numbers indicate the significant difference analyzed by Tukey’s HSD test (*p* < 0.05). The images were representative.

**Figure 5 cells-10-02884-f005:**
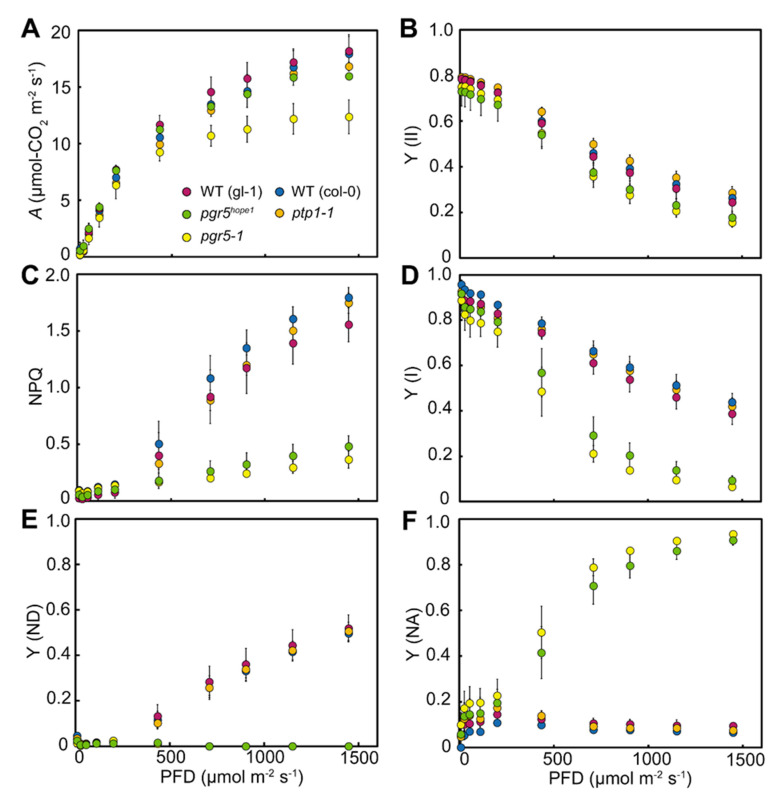
The photosynthetic capacity in *pgr5-1*, *pgr5^hope1^* and *ptp1-1*. Light intensity-dependent changes in photosynthetic parameters; (**A**), CO_2_ fixation rate, (**B**), the quantum yield in PSII (Y(II)), (**C**), non-photochemical quenching, (**D**), the quantum yield in PSI (Y(I)), (**E**), the ratio of oxidized P700 (P700^+^) in PSI (Y(ND)), (**F**), the ratio of excited P700 (P700*) in PSI (Y(NA)). Magenta, blue, green, yellow and orange are wild-type gl-1, wild-type Col-0, *pgr5^hope1^*, *pgr5-1* and *ptp1-1*, respectively. Data are means ± sd (*n* = 5). Experiments were independently repeated at least 3 times with similar results. Graphs show representative results. The results of statistical analysis of these data are summarized in the Appendix A.

**Figure 6 cells-10-02884-f006:**
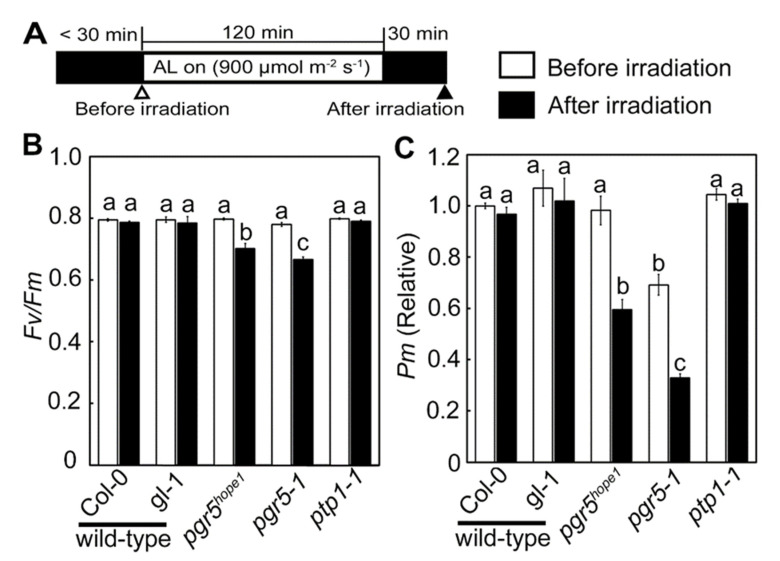
The extent of photoinhibition by constant intense light. (**A**) is an illustration of an experimental scheme for photoinhibition by constant intense light. The leaves that had been fully darkened (<30 min) were exposed to intense constant light (AL; 900 μmol-photons m^−2^ s^−1^) for 2 h under atmospheric conditions, and then darkened for 30 min. Saturated pulse flush was exposed before AL on (before irradiation) and after darkness for 30 min with AL off (after irradiation). (**B**,**C**) show the maximum quantum yield in PSII (*Fv*/*Fm*) and the maximum amount of photo-oxidizable PSI (*Pm*) before and after irradiation, respectively. Black and white bars are before and after irradiation. In (**C**), *Pm* is a relative value with the value wild-type Col-0 before irradiation being 1. Data are means ± sd. (*n* = 3) Experiments were independently repeated at least 3 times with similar results. Graphs show representative results.

**Table 1 cells-10-02884-t001:** The chlorophyll, leaf total N and Rubisco amount in *pgr5-1*, *pgr5^hope1^* and *ptp1-1*. After photosynthesis measurement (Figure 5), leaves were used for the determination of leaf components. Dara are means ± sd. (*n* = 4–5) Different alphabets beside the numbers indicate the significant difference analyzed by Tukey’s HSD test (*p* < 0.05).

Genotypes (Background)	Chlorophyll mmol m^−2^	Chlorophyll a/b	Leaf N mmol m^−2^	Rubisco g m^−2^
wild-type (Col-0)	0.377 ± 0.041 a	3.08 ± 0.20 a	80.6 ± 6.9 a	1.45 ± 0.17 a
wild-type (gl-1)	0.366 ± 0.024 a	3.09 ± 0.40 a	85.8 ± 4.6 a	1.42 ± 0.13 a
*pgr5^hope1^* (gl-1)	0.385 ± 0.009 a	2.96 ± 0.30 a	82.4 ± 5.6 a	1.48 ± 0.15 a
*pgr5-1* (gl-1)	0.287 ± 0.030 b	3.30 ± 0.20 a	78.4 ± 4.0 a	1.27 ± 0.09 a
*ptp1-1* (Col-0)	0.293 ± 0.019 b	3.29 ± 0.19 a	80.2 ± 5.6 a	1.51 ± 0.11 a

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
