# Peer review of "Identification of a Novel Mutation Exacerbated the PSI Photoinhibition in pgr5/pgrl1 Mutants; Caution for Overestimation of the Phenotypes in Arabidopsis pgr5-1 Mutant"

_cells, 2021, doi:10.3390/cells10112884_

Round 1

Reviewer 1 Report

The current manuscript revealed that pgr5-1 is a double mutant with exaggerated PSI photoinhibition. Notably, pgr5-1 and pgr5hope1 tended to cause PSI photoinhibition by excess light, yet, pgr5-1 exhibited an enhanced reduction in PSI activity. Introducing At2g17240, a candidate gene for the second mutation, into pgr5-1 restored the pale-green phenotype and partially restored PSI activity. The authors concluded that At2g17240, named PSI photoprotection 1 (PTP1), is involved in PSI photoprotection. This is an interesting manuscript, yet, open questions remain.

Other comments

  1. Suppl. Figure 1D, a loading control is missing. Is PGR5 indeed completely absent in the mutants? It is not clear, to which extent PGR5 can be detected by the antibody. These data should be flanked by mass spectrometric analyses.

  1. The same question holds true for the presence of At2g17240?

  1. Figure 2, A-C, it would be interesting to evaluate other electron transfer parameter such as (Y(ND) and Y(NA) in pgr5-hope1 and pgr5-hope1/PGR5 under such conditions, where PSI photoinhibition is not yet detectable.

  1. Fig.4, plant material was stemming from which conditions?

  1. Fig, 5, statistics are missing or are all presented data not significantly different to each other?

  1. Fig. S4A and Fig.5A, are pgr5-1 and pgrl1/ptp1 CO2 fixation rates significant different?

  1. PGRL1 is required for protecting PGR5 from degradation (Rühle et al., 2021). Fig. S4A and Fig.4A, the pgrl1/ptp1 phenotype seems to stronger as the pgr5. Could it be that PGR5 protein levels are further diminished in pgrl1/ptp1?

  1. pgr5 photo-sensitivity was also reported for Chlamydomonas pgr5 mutants (Johnson et al., 20149, yet, under conditions where PSI is not photo-inhibited, pgr5 mutants revealed increased Y(ND) and changes in b6f complex dependent Q cycle. Information, which could be interesting for discussion.

Author Response

Regarding the comments of Reviewer1,

1.) Figure 1D, a loading control is missing. Is PGR5 indeed completely absent in the mutants? It is not clear, to which extent PGR5 can be detected by the antibody. These data should be flanked by mass spectrometric analysis.

In the original paper of pgr5-1 (Munekage et al., 2002), They described that PGR5 was absent in the pgr5-1 mutant. Therefore, in theory, pgr5hope1 has the same point mutation and is not considered to be present in the PGR5 protein. In our experiment, the chemiluminescent detection reagent (SuperSignal West Dura extended duration Substrate, ThermoFisher) can be detected the mid-femtogram level protein. However, the mutation leads to amino acid substitution and reduces the stability of PGR5 protein. Thus, it is possible that PGR5 still exists at a very low level. Since there is currently no mass spectrometer in our laboratory, we cannot take immediate action on this point.

2.) The same question holds true for the presence of At2g17240.

So far, we have attempted to produce the PTP1(CGL20A) specific antibody which was against the artificial peptide sequence of PTP1, twice. And both antibodies could not detect the specific protein in the western blotting even in the wild-type plant.

However, the pgrl1ab ptp1-1 and pgr5-1 mutant showed very similar results in growth phenotypes (Figure2 and 3) and chlorophyll fluorescence parameters (Figure 5 and suppl. Figure S4). It has been confirmed that ptp1-1 (cgl20a) does not have transcripts of PTP1 by T-DNA insertion. Therefore, the ptp1-2 mutation of the pgr5-1 mutant is also considered to have a near-defective effect on PTP1 expression.

3.) Figure 2, A-C, it would be interesting to evaluate other electron transfer parameter such as (Y(ND) and Y(NA)) in pgr5-hope1 and pgr5hope1/PGR5 under such conditions, where PSI photoinhibition is not yet detectable.

We could not get the point of this question.

In figure 5, samples were from the stable light condition where PSI photoinhibition was not yet detectable in pgr5hope1.

Is this question means that comparison that the pgr5hope1 and PGR5 overexpression strains?

4.) Fig.4, plant material was stemming from which conditions?

In figure 4, leaf samples were from the constant light conditions. After measurement of photosynthetic parameters, leaves were sampled and applied to the western-blot (Figure 4) and determination of leaf components (Table1).

5.) Fig.5, statistics are missing or are all presented data not significantly different to each other?

We added the result of statistical analysis on the data in Fig. 5 as supplement table S3.

6.) Fig. S4A and Fig.5A, are pgr5-1 and pgrl1/ptp1 co2 fixation rates significant different?

The CO2 fixation rates of pgr5-1 and pgrl1/ptp1-1 were not significantly different.

7.) PGRL1 is required for protecting PGR5 from degradation (RÏ‹hle et al., 2021). Fig. S4A and Fig. 4A, the pgrl1/ptp1 phenotype seems to stronger as the pgr5. Could it be that PGR5 protein levels are further diminished in pgrl1/ptp1?

Again, the CO2 fixation rates of pgr5-1 and pgrl1/ptp1-1 were not significantly different. DalCorso et al., 2008 and RÏ‹hle et al., 2021 showed that PGR5 protein was diminished in pgrl1ab background. Thus, pgrl1ab/ptp1-1 mutant probably has less PGR5 protein, we didn’t detect the PGR5 protein in pgrl1ab/ptp1-1 mutant. However, we didn’t have any results to show the difference between pgr5-1 and pgrl1/prp1-1.

8.) pgr5 photosensitivity was also reported for Chlamydomonas pgr5 mutants (Johnson et al., 2014, yet, under conditions where PSI is not photo-inhibited, pgr5 mutants revealed increased Y(ND) and changes in b6f complex dependent Q cycle. Information, which could be interesting for discussion.

In our opinion, we don’t think that further discussion based on the paper Johnson et al 2014 is needed in our manuscript. 

Reviewer 2 Report

Dear authors,

Your research in my opinion is strongly interesting for the photosynthesis field, however some major concerns in regards mainly with the rationale of the existence of PTP1 (even is not clear why did you suspect it with At gene you mention in the manuscript) must be improved, as well as some aspects in the material and methods section. In my opinion the rest of your paper is OK for me. 

For specific comments please see the attached file.

Author Response

Regarding the comments of Reviewer2,

The manuscript was revised according to each comment.

Reviewer 3 Report

The manuscript describes the revelation of an additional mutation in the gene Atg17240 (named by the authors as PTP1) in the pgr5-1 Arabidopsis mutant. The authors state that the presence of this mutation, responsible for the pale-green phenotype of pgr5-1 mutant,  might lead to overestimation of the true degree of PSI photoinhibition due to the absence of PGR5 protein in Arabidopsis plants. This result is important as the pgr5-1 mutant is often used in studies on the regulation mechanisms involving cyclic electron transport around PSI. The manuscript is well written and reflects deep knowledge of the Miyake’s group in this research area. I have several remarks to the text, which should be sorted out before publication of these results.

  • The presented photosynthetic function characteristics (Figure 5) of the mutants describe only so called “steady state” and they did not differ between prg5-1 and pgr5-hope1 (NPQ, Y(I), Y(II),Y(NA), Y(ND)). It is important to know whether there any differences in parameters during chlorophyll fluorescence induction (before the “steady state” is reached). I do not expect changes of the parameters, but it should be checked at least for selected actinic light intensities. The potential changes in these parameters between pgr5-1 and pgr5-hope1 would open many other questions on the function of PTP1.
  • It is strange to me that you were not able to detect PTP1 protein in WT or in the mutants where it should be present (lines 297-8). What kind of methods did you use? Did you use mass spectrometry? This issue should be discussed.
  • Table 1: There are no statistically significant changes in the chlorophyll a/b ratio between variants, thus, you should not write that “… chlorophyll a/b ratio tended to increase in pgr5-1 and ptp1-1 …”
  • In Figure 4 you present protein densities for representative samples with their relative quantities for this single experiment. In the legend to this figure, you have written that you performed this experiment three times. Thus, the quantities +-SD should be presented in this figure. The protein content changes between the variants are discussed in the text, are they statistically significant?

Author Response

Regarding the comments of Reviewer3,

1.) It is important to know whether there are any differences in parameters during chlorophyll fluorescence induction (before the “steady state” is reached). I do not expect changes of the parameters, but it should be checked at least for selected actinic light intensities…

So far, no difference has been observed in the chlorophyll fluorescence parameters of pgr5-1 and pgr5hope1 even in the induction phase of photosynthesis. In case adding them to the manuscript, the data is currently insufficient, so it would take about one month to revise them.

2.) It is strange to me that you were not able to detect PTP1 protein in WT or in the mutants where it should be present (line 297-8). What kind of methods did you use? Did you use mass spectrometry?

So far, we have attempted to produce the PTP1(CGL20A) specific antibody which was against the artificial peptide sequence of PTP1, twice. And both antibodies could not detect the specific protein in the western blotting even in the wild-type plant. We usually use the chemiluminescence detection method (detection reagent can be detected up to mid-femtogram level.) for western blotting and do not have mass spectrometry.

3.) Table1: There are no statistically significant changed in the chlorophyll a/b ratio between variants, thus, you should not write that “… chlorophyll a/b ratio tended to increase in pgr5-1 and ptp1-1…”

We revised the manuscript.

4.) In figure 4 you present protein densities for representative samples…

We revised Figure 4 with sd values and the result of statistical analysis.

Round 2

Reviewer 1 Report

The manuscript has improved, most aspects have been addressed. However, a few questions remain open.

If no MS setup is available, an immuno-titration experiment should be done to estimate the amounts of PGR5 in pgr5 plants versus WT. This also accounts for pgr5-hope1, pgr5-hope1/PGR5 and in pgrl1/ptp1.

The availability of pgr5-hope1 and pgr5-hope1/PGR5 pgr5 allows for more thorough electron transfer analyses as pgr5-hope1 shows less PSI photosensitivity.  

Author Response

Since the activity of PGR5 protein cannot be measured, immuno-titration experiments cannot be performed. Therefore, the result of relative quantification by immunoblotting was added to the manuscript instead. However, in the pgr5-1, pgr5hope1, pgrl1ab, pgrl1ab ptp1 mutants, the PGR5 protein could not be detected even when the maximum amount of sample was applied.

Reviewer 2 Report

Adequate addressing of muy comments. The paper must be accepted in its present form.

Author Response

Thank you for your valuable comments.  We were able to revise the manuscript better.